# Heat-Killed *Staphylococcus aureus* Induces Bone Mass Loss through Telomere Erosion

**DOI:** 10.3390/ijms24043179

**Published:** 2023-02-06

**Authors:** Songyun Deng, Mankai Yang, Jianwen Su, Naiqian Cui, Siyuan Wu, Guangyan Zhang, Lei Wang, Yilong Hou, Yu Chai, Bin Yu

**Affiliations:** 1Division of Orthopaedic Surgery, Department of Orthopaedics, Nanfang Hospital, Southern Medical University, Guangzhou 510515, China; 2Guangdong Provincial Key Laboratory of Bone and Cartilage Regenerative Medicine, Nanfang Hospital, Southern Medical University, Guangzhou 510515, China

**Keywords:** telomere, senescence, bone loss, heat-killed *S. aureus* (HKSA), chronic inflammation

## Abstract

The mechanism of systemic osteoporosis caused by chronic infection is not completely clear, and there is a lack of reasonable interventions for this disease. In this study, heat-killed *S. aureus* (HKSA) was applied to simulate the inflammation caused by the typical clinical pathogen and to explore the mechanism of systemic bone loss caused by it. In this study, we found that the systemic application of HKSA caused bone loss in mice. Further exploration found that HKSA caused cellular senescence, telomere length shortening, and telomere dysfunction-induced foci (TIF) in limb bones. As a well-known telomerase activator, cycloastragenol (CAG) significantly alleviated HKSA-induced telomere erosion and bone loss. These results suggested that telomere erosion in bone marrow cells is a possible mechanism of HKSA-induced bone loss. CAG may protect against HKSA-induced bone loss by alleviating telomere erosion in bone marrow cells.

## 1. Introduction

Chronic inflammation has been reported to cause bone loss [1,2], while the underlying mechanism is not completely clear. Cellular senescence of osteogenesis cells was found to be an important cause of bone loss [3]. In a previous study, we found that patients with chronic osteomyelitis exhibit systemic telomere shortening [4]. Therefore, it is reasonable to hypothesize that telomere shortening, an established mechanism of cellular senescence [5,6], may involve bone loss caused by infection-related chronic inflammation.

Telomeres are sequences of repetitive DNA that cover the end of chromosomes to prevent degradation or recombination and maintain genomic stability [7]. Due to the inability of the replication machinery to fully replicate linear DNA molecules, telomeres shorten with each cellular division, thereby limiting the lifespan of normal somatic cells [8]. When telomeres become critically short and cannot protect the chromosomes, DNA repair proteins and signaling molecules such as γH2AX and 53BP1 at the site of telomere erosion occurs as the telomere dysfunction-induced foci (TIFs) [9,10]. There is considerable evidence to show that a reduction in the length of telomeres is associated with the failure of cell division and senescence of normal cells [5,6,11], and that chronic inflammation can contribute to the rate of attrition of telomere length [12].

*S. aureus* is a common cause of chronic inflammation and is known to contribute to the development of various chronic inflammatory conditions such as osteomyelitis, chronic liver abscess, and septic arthritis [13,14,15]. In this study, we constructed a systemic chronic inflammation model using heat-killed *S. aureus* (HKSA), as HKSA is a well-established model of chronic inflammation, such as lung injury, enterocyte effacement, intestinal epithelium destruction, allowing us to mimic persistent inflammation without the risk of additional infections [16,17,18,19]. The objective of this study was to investigate the role of telomere erosion in HKSA-induced bone loss.

## 2. Results

### 2.1. HKSA Decreased Bone Volume in Mice

To determine whether the application of HKSA induces bone loss, we used μCT to measure the architecture of the femoral bone. A significant reduction in the mass of trabecular bone was observed in the experiment mice after 6 weeks of HKSA treatment (Figure 1A). The HKSA-treated mice exhibited reduced trabecular bone volume, Bone Volume/Tissue Volume (BV/TV) (10.33 ± 1.13% vs. 5.95 ± 0.38%, *p* < 0.05) (Figure 1B), Trabecular Thickness (Tb. Th) (39.83 ± 0.49 μm vs. 33.98 ± 0.76 μm, *p* < 0.001) (Figure 1D) and Bone Mineral Density (BMD) (652.46 ± 5.97 mg HA/ ccm vs. 623.64 ± 10.02 mg HA/ ccm, *p* < 0.05) (Figure 1G), a greater Structure Model Index (SMI) (2.53 ± 0.11 vs. 3.02 ± 0.05, *p* < 0.01) (Figure 1C), and increased Trabecular Separation (Tb. Sp) (0.20 ± 0.009 vs. 0.23 ± 0.006, *p* < 0.05), compared with the control littermates (Figure 1F). H&E staining also showed that the number and the thickness of trabecular bones were significantly decreased in the tibia of HKSA mice (Figure 1H). Together, these data revealed that HKSA has adverse effects on bone architecture.

Since the cause of bone loss is an imbalance in the osteogenesis-osteoclastic relationship, we further examined osteogenesis and osteoclast-related indicators. As osteocalcin (OCN) is a marker of bone formation and it is considered as one of the most sensitive markers of bone remodeling, we conducted immunohistochemical staining of OCN to assess changes in osteogenic function. Tartrate-resistant acid phosphatase (TRAP) is produced by osteoclasts and is used as a marker of osteoclast activity and bone resorption, we performed TRAP staining to assess changes in bone resorption function. The HKSA-treated mice exhibited a significantly reduced number of OCN^+^ cells (21.22 ± 2.69/mm vs. 14.1 ± 0.68/mm, *p* < 0.05) (Figure 1I,J), while no significant change in the number of TRAP^+^ cells (Figure 1K,L) was observed on the surface of trabecular bone, which suggested that the change in osteogenic activity mainly contributed to osteoporosis caused by HKSA.

### 2.2. HKSA Induced Cellular Senescence in Bone

To determine whether the application of HKSA induces cellular senescence, the activity of the enzyme beta-galactosidase, which is present in senescent cells but not in normal cells, was assessed using the SA-β-gal staining in the HKSA and control femurs. Compared with control mice, a significant increase in SA-β-gal^+^ cells were found in HKSA mice (13.8 ± 2.66% vs. 22.95 ± 1.79%, *p* < 0.05) (Figure 2A,B).

### 2.3. HKSA Caused Telomere Erosion in Cells of Bone

To evaluate telomere erosion caused by HKSA, we compared the HKSA-treated mice with control mice for the TIFs. We used γ-H2A and Cy3-Tel to define TIFs, and found a significant increase of TIF^+^ cells in HKSA-treated mice than in control mice (12.60 ± 1.20% vs. 28.71 ± 1.48%, *p* < 0.001) (Figure 3A,B).

To further explore the telomere length erosion caused by HKSA, we compared the cell telomere length between HKSA mice and control mice using qPCR. We found that the telomere length in the femur of HKSA mice was significantly shorter than that of control mice (fold change = 0.50, *p* < 0.05) (Figure 3C).

### 2.4. CAG Protected HKSA-Treated Mice from Telomere Erosion in Cells of Bone

Telomerase activation is the most common method for protecting telomeres, and CAG, a molecule that is derived from the Astragalus plant, is the most recognized telomerase-activator [20,21]. We investigated whether telomere lengthening can improve HKSA-associated osteoporosis by using CAG. We found that the application of CAG ameliorated HKSA-induced telomere shortening in femur cells (HKSA vs. HKSA + CAG = 0.42 vs. 0.95, *p* < 0.05) (Figure 4B).

Then, we explored the effect of CAG on TIFs (Figure 4A). Compared with HKSA-treated mice, CAG application in HKSA-treated mice significantly reduced TIF^+^ cells around trabecular bone (HKSA vs. HKSA + CAG = 25.99 ± 2.56% vs. 17.26 ± 1.71%, *p* < 0.05) (Figure 4C).

### 2.5. CAG Protected HKSA-Treated Mice from Cellular Senescence

Compared with HKSA mice, the rate of SA-β-gal^+^ cells were significantly reduced around trabecular bone in HKSA + CAG mice (Figure 5A,B) (HKSA vs. HKSA + CAG = 25.72 ± 0.69% vs. 20.73 ± 0.63%, *p* < 0.05). The SA-β-gal results suggested that CAG application alleviates cellular senescence induced by HKSA.

### 2.6. CAG Protected HKSA-Treated Mice from Bone Loss

Micro-CT was used to measure the architecture of the femoral bones. A significant increase in the mass of trabecular bone was observed in HKSA + CAG mice compared with HKSA mice (Figure 6A). The HKSA + CAG treated mice exhibited increased BV/TV (HKSA vs. HKSA + CAG = 8.50 ± 0.57% vs. 13.13 ± 0.47%, *p* < 0.05) (Figure 6B), Tb. Th (HKSA vs. HKSA + CAG = 35.77 ± 0.86 μm vs. 39.91 ± 0.11 μm, *p* < 0.05) (Figure 6D), decreased SMI (HKSA vs. HKSA + CAG = 3.13 ± 0.05 vs. 2.65 ± 0.70, *p* < 0.05) (Figure 6C), and decreased Tb. Sp (HKSA vs. HKSA + CAG = 260.34 ± 14.7 μm vs. 212.44 ± 10.75 μm, *p* < 0.05) (Figure 6F) compared with HKSA-treated littermates.

The HKSA + CAG treated mice exhibited increased OCN positive cells (HKSA vs. HKSA + CAG = 12.50 ± 0.97/mm vs. 19.90 ± 0.64/mm, *p* < 0.05) (Figure 6H,I) compared with HKSA-treated littermates, while no significant change in the number of TRAP^+^ cells (Figure 6J,K) on the surface of trabecular bone, which suggested that CAG protects against bone loss caused by HKSA mainly by improving osteogenic function. All these data suggested that CAG can alleviate the bone loss caused by HKSA.

## 3. Discussion

Our findings suggested that HKSA causes osteoporosis by causing telomere shortening, which provides a new explanation for osteoporosis caused by chronic inflammation. We found that the telomerase activator CAG can alleviate HKSA-induced telomere damage and bone loss, which may provide a new treatment for osteoporosis caused by chronic inflammation.

In this study, we found that chronic HKSA application could induce osteoporosis. As a causative substance of *Staphylococcus aureus*, the most common pathogenic bacteria in clinical practice, HKSA is often used as an inflammatory inducer [16,17,18,19]. There is considerable evidence to show that chronic inflammation can lead to osteoporosis by inhibiting the osteogenic activity of osteogenic cells [22,23,24] or by promoting the bone resorptive activity of osteoclasts [23,24,25]. This study found that HKSA caused an increase in the number of OCN^+^ cells, suggesting that HKSA may mainly lead to bone loss by inhibiting the osteogenic activity of osteogenic cells.

This study found, for the first time, that systemic application of HKSA can cause telomere shortening in bone marrow cells in vivo. HKSA can cause telomere loss for a variety of reasons. One reason is that through replication stress [6,7]. HKSA, as an immunogen, activates the immune system, leading to massive replication and expansion of immune cells, which in turn leads to telomere shortening through terminal replication effects [26]. Replicating telomere shortening is generally prolonged by telomerase, which most somatic cells lack [27,28]. The other cause maybe the chronic oxidative stress caused by HKSA [29,30], oxidative stress causes telomere damage by directly attacking the guanine triad of telomeres, or by attacking telomere protective complexes shelterin, which contains TRF1, TRF2, RAP1, POT1, TIN2, and TPP1, resulting in loss of telomere protection [9,31,32].

In order to further confirm that telomere erosion is the mechanism of bone loss caused by HKSA, the most recognized telomere protector, CAG [33], was used to rescue telomere damage caused by HKSA. The results showed that the telomere length of HKSA + CAG mice was significantly longer than that of HKSA mice, and the TIF signal was the opposite, suggesting that CAG improved HKSA-induced telomere damage. Compared with HKSA mice, the bone masses of HKSA + CAG mice were significantly increased, proving that telomere erosion is one of the mechanisms of bone loss caused by HKSA. Compared with control mice, CAG mice had longer telomeres, but they did not extend them significantly, and TIF signaling did not improve significantly, indicating that CAG did not significantly improve telomeres in bone marrow cells under physiological conditions. In addition, as a recognized telomerase activator, CAG increased telomere length in HKSA mice while also decreasing TIF signaling. CAG may reduce the number of very short telomeres and improve telomere TIF signaling by lengthening telomeres. In addition, previous studies have found that telomerase can improve chronic oxidative stress [34,35], therefore, CAG may also improve TIF by reducing chronic oxidative stress.

There were several limitations in this study. Firstly, the cell types that cause bone loss from HKSA are unclear. Inflammatory response and telomere erosion in immune cells may be one of the indirect mechanisms of bone loss caused by HKSA, telomere erosion in osteogenic cells may be one of the direct mechanisms, and more research is needed to clarify the cellular mechanism. Secondly, it is unclear whether HKSA shortens telomeres by inhibiting telomerase activity, promoting replication aging, or by oxidative pressure-induced telomere gene damage or shelterin protein damage. More research is needed for the future. Thirdly, although the change in mean telomere length and the change in TIF signal are synchronized in this study, it does not indicate that the two are causally related. Many previous studies have found that only very short telomeres directly cause TIF signaling [36,37]. In this study, the two may be caused by different mechanisms, which must be further distinguished in future studies.

## 4. Materials and Methods

### 4.1. Bacterial Strains and Preparation of HKSA

*S. aureus* was isolated from osteomyelitis patients from the Department of Orthopedics, Nanfang Hospital, Southern Medical University. The strain of *Staphylococcus aureus* was verified by the Department of Laboratory Medicine, Nanfang Hospital, Southern Medical University. An isolated colony of *S. aureus* from a fresh tryptic soy agar plate was inoculated in 10 mL of fresh tryptic soy broth overnight at 37 °C with shaking at 200 revolutions/min (rpm). Bacteria were harvested by centrifugation, washed twice with phosphate-buffered saline (PBS), and resuspended in PBS. The concentration of the *S. aureus* strain was adjusted to 5 × 10^8^ colony-forming units/mL (CFUs/mL) according to an optical density (OD) of 600 nm by a microplate reader (SpectraMax i3x, Molecular Devices, Urstein, Austria) and then heated at 80 °C for 30 min [18,19] and subsequently stored at −20 °C.

### 4.2. Animals and Experiment Protocol

Protocols for animal experiments were conducted following the Guide for the Care and Use of Laboratory Animals of Nanfang Hospital, Southern Medical University. Pathogen-free mice were housed under standard conditions in a 12 h light/dark cycle at 20–25 °C and were fed with standard laboratory chow. Eight to ten-week-old (22–26 g) C57BL/6 male mice were used for the animal models. The dosing regimen was derived from a previous study [38]. Briefly, the HKSA group mice were treated by intraperitoneal injection of 200 µL HKSA, containing approximately 1 × 10^8^ CFU, and the control group mice were injected with an equal volume of sterile normal saline (NS) every other day at a consistent time. The treatment lasted for six weeks. To lengthen the telomere of the mice, we used CAG (MCE, Shanghai, China), following the previous studies [20,39]. CAG was dissolved in DMSO and then diluted with 2% Tween 80 in NS, ensuring the final concentration of DMSO was less than 5%. The CAG-treated group mice were intraperitoneally injected with 200 µL NS containing 20 mg/kg CAG, the HKSA + CAG group mice were injected with 200 µL NS containing 1 × 10^8^ CFU HKSA and 20 mg/kg CAG, the control group mice were injected with 200 µL NS containing an equal amount of DMSO and Tween 80, the HKSA group mice were injected with 200 µL NS containing 1 × 10^8^ CFU HKSA and an equal amount of DMSO and Tween 80. All the mice were injected every other day at a consistent time, for six weeks.

### 4.3. Micro-Computed Tomography (μCT) Analysis

Femora from the model mice were dissected free of the soft tissue, fixed in 4% paraformaldehyde, and stored in 70% ethanol, before being imaged using a μCT specimen scanner (Scanco Medical, AG, Bern, Switzerland). The scan was performed using an X-ray energy of 55 kVp and a current of 145 mA, with a voxel size of 12 μm and an integration time of 400 ms. Cross-sectional images of the distal femur were used to perform a three-dimensional histomorphometric analysis of the trabecular bone. Images were scanned from the distal end of the femur, corresponding to a 0.215 mm to 1.94 mm area above the growth plate. Quantitative analyses were carried out using IPL software (Image Processing Language V5.15, Scanco Medical AG, Switzerland). BV/TV, Tb. Th, BMD, SMI, Tb. Sp and Tb. N are all parameters that are used to describe the microarchitecture and mineralization of bone tissue. BV/TV is used to describe the ratio of bone tissue to total tissue, Tb. Th is used to describe the average thickness of the trabecular bone tissue, BMD is used to describe the amount of mineral content in a given volume of bone tissue, SMI is used to describe the degree of anisotropy (directional dependence) of the trabecular bone tissue and is positively correlated with the degree of osteoporosis, and Tb. N is used to describe the number of trabeculae per given volume.

### 4.4. Histochemistry

Tibiae on one side of the model mice were fixed in 4% paraformaldehyde and decalcified in 0.5 M ethylenediaminetetraacetic acid (pH 8.0), then embedded in paraffin. The contralateral tibiae were fixed and decalcified while embedded in an optimal cutting temperature compound (OCT) for freezing. Continuous longitudinal sections of 4 μm thickness were cut for paraffin-embedded samples, and 10 μm thick continuous longitudinal sections were cut for frozen samples. For H&E staining, the previously obtained paraffin sections were preheated in an air oven at 60 °C and then dewaxed and rehydrated in xylene and ethanol solution (reduced concentration by 100–70%). The sections were then successively soaked in H&E dyes. After dehydration with graded alcohol, the histological observations were taken with a microscope (BX63, Olympus, Tokyo, Japan).

For immunohistochemistry staining, osteocalcin (OCN) staining was performed on the deparaffinized and rehydrated sections using the rabbit anti-osteocalcin (OCN) (ab93876, Abcam, Shanghai, China) and VECTASTAIN ABC HRP kit (Cat. PK-4001, Vector Laboratories, Burlingame, CA, USA) according to the manufacturer’s protocol. Peroxidase activities of sections were revealed with 3,3′-diaminobenzidine substrate (Vector Laboratories). Images were acquired with a microscope (BX63, Olympus, Japan). The number of OCN^+^ cells per millimeter of bone surface (N. OCN^+^/mm) was quantified.

For detecting the osteoclastic activity, TRAP staining was performed on the deparaffinized and rehydrated sections using a leukocyte acid phosphatase kit (Cat. 387A-1, Sigma-Aldrich, Darmstadt, Germany) according to the manufacturer’s directions. Images were acquired with a microscope (BX63, Olympus, Japan). The number of TRAP^+^ cells per millimeter of bone surface (N. of TRAP^+^/mm) was quantified.

### 4.5. SA-β-Galactosidase Staining

The contralateral tibiae were fixed and decalcified while embedded in an optimal cutting temperature compound (OCT) for freezing. As a marker to identify senescent cells, an SA-β-galactosidase (SA-β-gal) assay was performed on the aforementioned frozen slices using a standard staining kit (Cell Signaling Technology, Beverly, MA, USA; Catalog# 9860). Briefly, frozen bone sections were air-dried, washed in PBS before and after fixation, and incubated with the β-gal staining solution containing the substrate X-gal as per the manufacturer’s instructions. The slides were incubated at 37 °C for 16 to 20 h, and the observations were taken with a microscope 20X objective to take SA-β-gal figures. Blue stained cells were defined as SA-β-gal^+^ cells, SA-β-gal^+^ cells lining the bone surface were quantified per mm of trabecular bone surface.

### 4.6. Quantification of TIFs

TIFs were calculated to evaluate the DNA damage caused by telomere erosion and was reflected by Immuno-FISH. As described [40], formalin-fixed paraffin-embedded bone slices were successively hydrated by 100%, 95%, and 70% methanol, along with distilled water, for 5 min. For antigen retrieval, the slides were heated in Tri-EDTA buffer (10 mM Tris +1 mM EDTA +0.1% Triton-X100) to 70 °C for 1 h. After cooling to room temperature, the slides were rinsed with PBST (PBS with 0.1% Tween 20) buffer twice for 5 min. After blocking in 10% donkey serum in PBST, slices were incubated with the primer antibody, rabbit γ-H2A.X (Abmart, China, 1:400), at 4 °C overnight. The slices were then washed three times in PBS for 5 min, and incubated with DyLight 488-conjugated secondary antibody (A23220, Abbikine, Beijing, China, 1:200) for 30 min. Following incubation, slides were washed three times in PBS for 5 min, dehydrated with 70%, 85%, and 100% ethanol for 1 min each, and dried in air for 1 min. The slices were denatured at 85 °C for 10 min in hybridization buffer (60% formamide, 20 mM Na2HPO4, 20 mM Tris pH 7.4, 2 x SSC and 0.1 ug/mL salmon sperm DNA) containing 200 nM Cy3-labelled telomere specific (CCCTAA) peptide nucleic acid probe (F1002, Beijing, China, Panagene), followed by hybridization for 1 h at room temperature in the dark. The slices were washed twice with wash buffer (2 x SSC/0.1% Tween-20) for 10 min at 55–60 °C, and then at room temperature for 1 min. Then, the sections were incubated with the DNA-specific fluorescent stain 4′,6-diamidino-2-phenylindole (DAPI) to image the nucleus, then mounted and imaged with an Axio Imager.D2 fluorescence microscope (Carl Zeiss, White Plains, NY, USA) at a resolution of 100×. A minimum of two TIFs were defined as TIF^+^ cells.

### 4.7. Telomere Length Measurement

After grinding the bones in liquid nitrogen, the DNA of bone tissue was extracted using a TIANamp Genomic DNA Kit (DP304, TIANGEN, Beijing, China) and stored at –20 °C. The telomere length was analyzed with a quantitative real-time PCR that compared the telomere repeat sequence copy numbers (T) to a reference single copy-gene copy number (S), 36b4 [41].

Each 25 µL PCR reaction mixture contained 2.5 ng of DNA, 7.5 µL of Power SYBR Green PCR master mix (Accurate Biology, Changsha, China), 0.5 µL of rox (Accurate Biology, China), and 150 nmol/l of telomere-specific primers or 100 nmol/l 36b4 primers. The PCR program for the telomeric amplification was as follows: a 95 °C activation step, followed by 40 cycles of 95 °C, 15 s, and 58 °C, 60 s. The standard curve was prepared using serial dilutions of a DNA sample of a known quantity. The primer sequences used in the qPCR were, for telomeres (forward: CGGTTTGTTTGGGTTTGGGTTTGGGTTTGGGTTTGGGTT; reverse: GGCTTGCCTTA CCCTTACCCTTACCCTTACCCTTACCCT), and for 36b4 (forward CACACTCCATCATCAATGGGTACA; reverse: CAGCAAGTGGGAAGGTGTACTCA) [42]. The relative telomere length was calculated using the threshold cycle (2−ΔΔCT) method, with 36b4 for normalization.

### 4.8. Statistics

The data and statistical analysis were consistent with the pharmacological recommendations on experimental design and analysis. All the quantitative data were expressed as a mean ± S.E., and an independent Student *t*-test was used to compare the two groups. For multiple comparisons, one-way analysis of variance (ANOVA) with the Bonferroni post-test was used. Statistical analysis was performed using SPSS 20.0 (International Business Machines Corporation, IBM Corp., Armonk, NY, USA). The significance level was defined as *p* < 0.05.

## 5. Conclusions

In conclusion, this study found that telomere erosion in bone marrow cells is a possible mechanism of HKSA-induced bone loss. CAG may protect against HKSA-induced bone loss by alleviating telomere erosion in bone marrow cells. Thus, our data suggest that telomere protection may be an important therapeutic for chronic inflammation-induced osteoporosis.

## Figures and Tables

**Figure 1 ijms-24-03179-f001:**
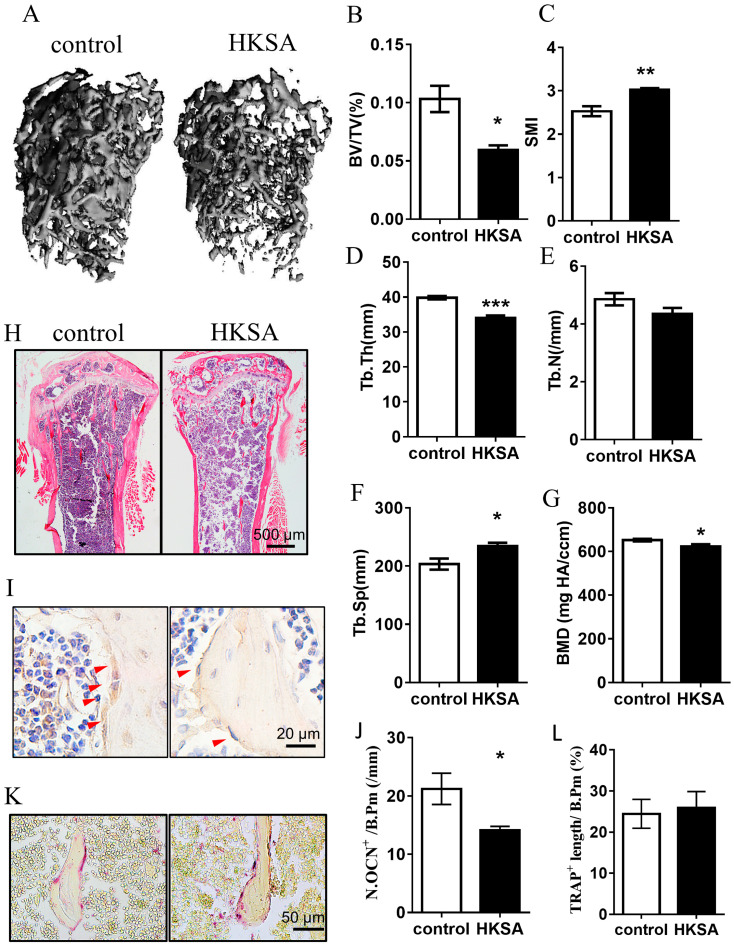
HKSA caused bone loss in mice. (**A**–**G**) Micro-CT scanning analysis of the femur from the control (*n* = 4 animals) and HKSA-treated mice (*n* = 4 animals). (**A**) Three-dimensional images obtained by micro-CT. (**B**) Bone volume fraction BV/TV. (**C**) Structural model index (SMI). (**D**) Trabecular thickness Tb. Th. (**E**) The number of trabeculae Tb. N. (**F**) Trabecular separation degree Tb. Sp. (**G**) Bone mineral density, BMD, of distal femoral metaphyseal. (**H**) Control and HKSA mice tibiae were harvested and processed for HE staining; the magnification is 4X, and the scale is 500 μm. (**I**) Representative images of immunohistochemistry staining of osteocalcin (OCN), red arrows indicate the OCN^+^ cells, the scale is 20 μm. (**J**) Quantification of OCN^+^ cells per bone surface (N. OCN^+^/mm) (*n* = 4 animals). (**K**) Representative images of tartrate-resistant acid phosphatase (TRAP) staining. Scale bars represent 50 μm. (**L**) Quantitative analysis of TRAP^+^ length per bone surface (*n* = 4 animals). All data were analyzed by two independent sample *t*-tests and expressed as the mean ± S.E. * *p* < 0.05 compared with the control group, ** *p* < 0.01 compared with the control group, *** *p* < 0.001 compared with the control group.

**Figure 2 ijms-24-03179-f002:**
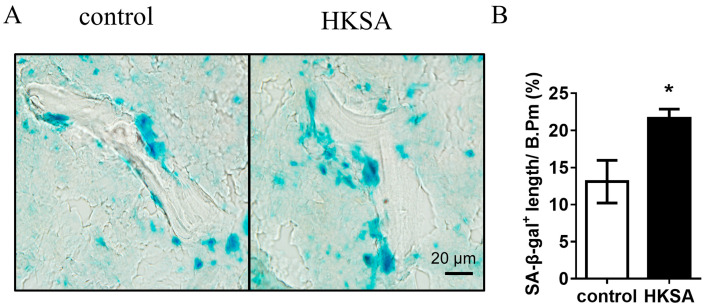
HKSA causes senescence of bone marrow cells. (**A**) SA-β-gal staining shows the presence of senescent cells in frozen tibial sections of control (*n* = 4 animals) and HKSA-treated mice (*n* = 4 animals). (**B**) Percent of SA-β-gal^+^ length lining the bone surface. All data were analyzed by two independent sample *t*-tests and were expressed as the mean ± S.E. * *p* < 0.05 compared with the control group.

**Figure 3 ijms-24-03179-f003:**
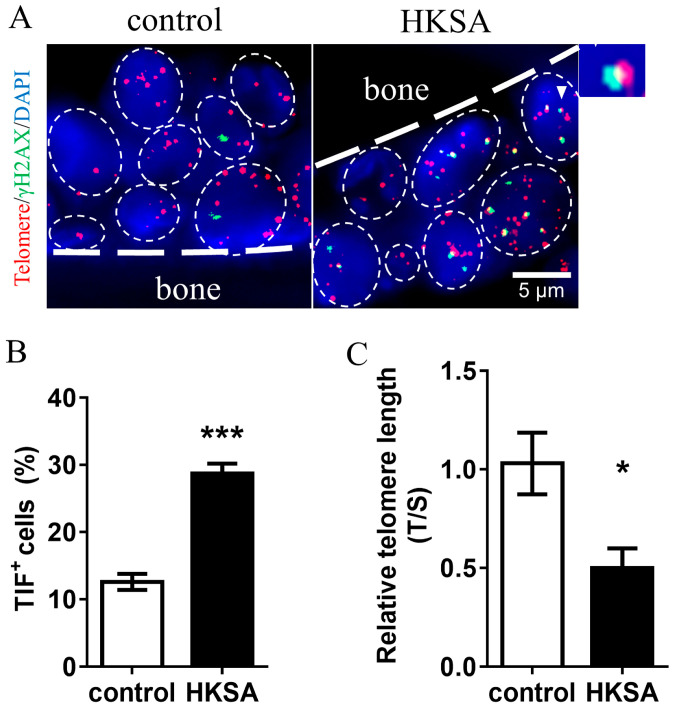
HKSA causes telomere erosion of bone marrow cells. (**A**) Control (*n* = 4 animals) and HKSA (*n* = 4 animals) mice tibiae were harvested and processed for TIFs detection. The co-localizations of telomeres (red) and γH2AX (green) were defined as TIF foci, and a minimum of two TIFs were defined as a TIF^+^ cell. The white triangle shows images representative of TIF foci. (**B**) Percentage of TIF^+^ cells around the trabecular bone. (**C**) Control (*n* = 4 animals) and HKSA (*n* = 4 animals) mice femurs were harvested and processed for telomere length detection. The relative telomere length compared to the reference single copy-gene (T/S) was detected by qPCR. All data were analyzed by two independent sample *t*-tests and expressed as the mean ± S.E. * *p* < 0.05 compared with the control group, *** *p* < 0.001 compared with the control group.

**Figure 4 ijms-24-03179-f004:**
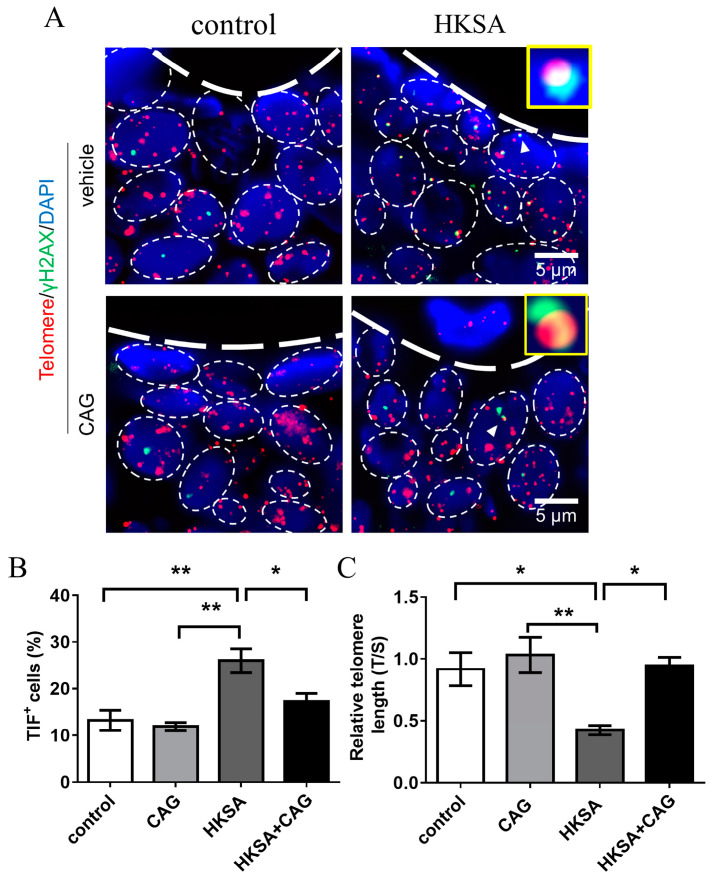
CAG ameliorates HKSA-induced telomere erosion. (**A**) Control, CAG, HKSA, and HKSA + CAG (*n* = 4 animals in all groups) mice tibiae were harvested and processed for TIFs detection. The co-localizations of telomeres (red) and γH2AX (green) were defined as TIF foci, and a minimum of two TIFs cells were defined as a TIF^+^ cell. The white triangle shows images representative of TIF foci. (**B**) Percentage of TIF^+^ cells around the trabecular bones. (**C**) Control, CAG, HKSA, and HKSA + CAG (*n* = 4 animals in all groups) mice femurs were harvested and processed for telomere length detection. The relative telomere length compared to the reference single copy-gene (T/S) was detected by qPCR. Quantitative values are presented as the mean ± S.E. One-way ANOVA was used to conduct multiple comparisons, and the Bonferroni test was used to identify where the differences lay. * *p* < 0.05 compared with the HKSA group, ** *p* < 0.01 compared with the HKSA group.

**Figure 5 ijms-24-03179-f005:**
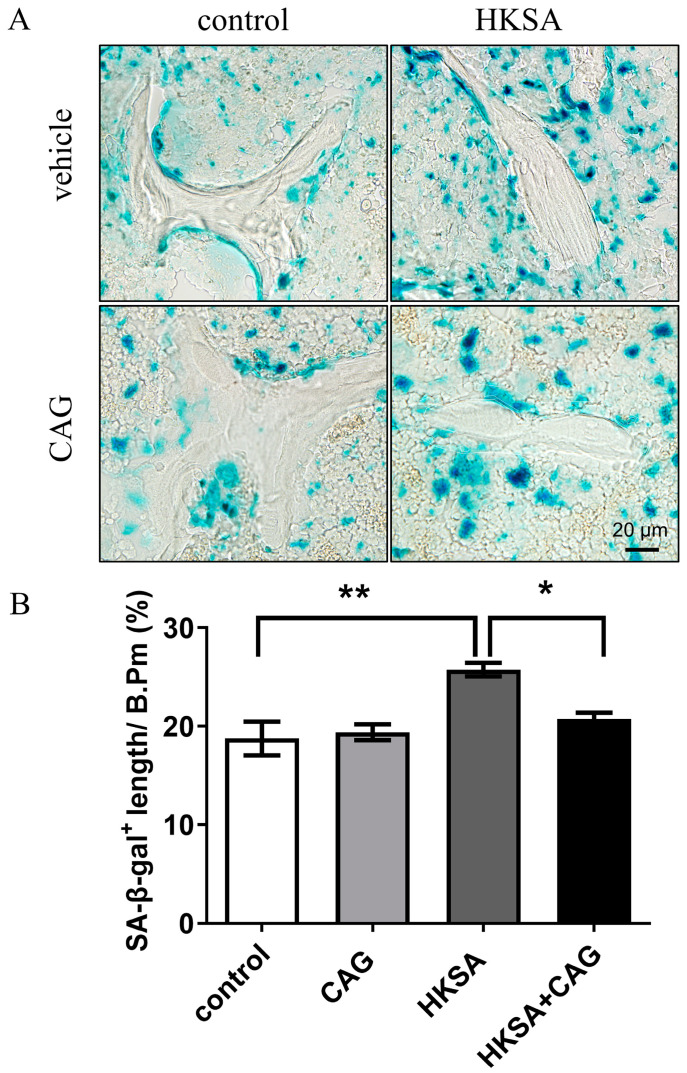
CAG ameliorates HKSA-induced cellular senescence. (**A**) SA-β-gal staining shows the presence of senescent cells in frozen tibial sections of control, CAG, HKSA, and HKSA + CAG (*n* = 4 animals in all groups) mice. (**B**) Percent of SA-β-gal^+^ length lining the trabecular bone surface. Quantitative values are presented as the mean ± S.E. One-way ANOVA was used to conduct multiple comparisons, and the Bonferroni test was used to identify where the differences lay. * *p* < 0.05 compared with the HKSA group, ** *p* < 0.01 compared with the HKSA group.

**Figure 6 ijms-24-03179-f006:**
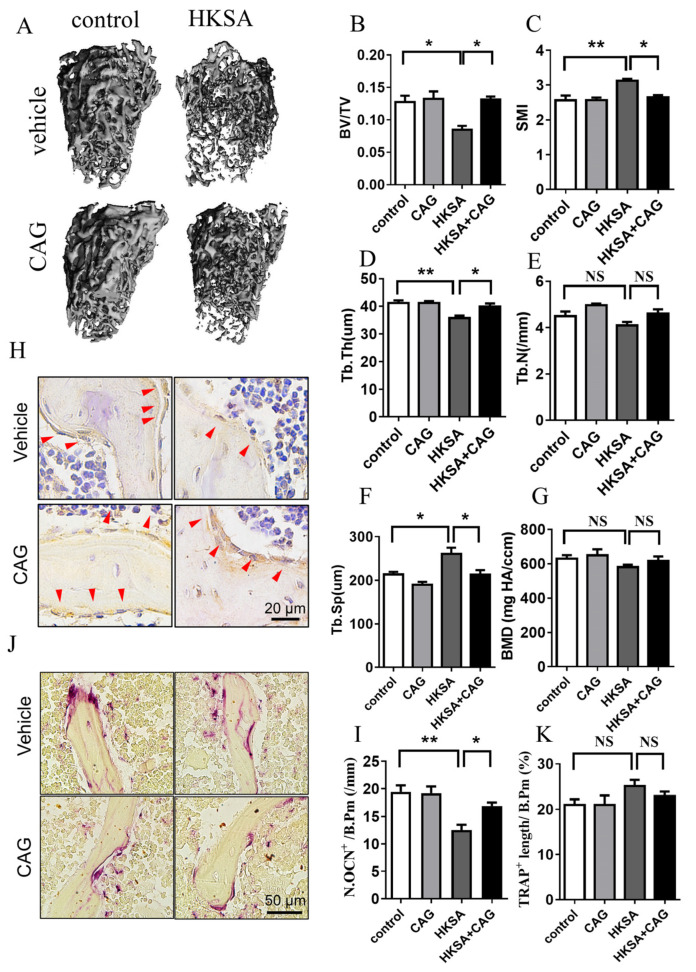
CAG ameliorates HKSA-induced bone loss. Micro-CT scanning analysis of control, CAG, HKSA, and HKSA + CAG (*n* = 4 animals in all groups) mice. (**A**) 3D images obtained by micro-CT. (**B**) Bone volume fraction BV/TV. (**C**) SMI. (**D**) Tb.Th. (**E**) Tb.N. (**F**) Tb.Sp. (**G**) BMD of distal femoral metaphyseal. (**H**) Representative images of immunohistochemistry staining of OCN, red arrows indicate the OCN^+^ cells, the scale is 20 μm. (**I**) Quantification of OCN^+^ cells per bone surface (N. OCN^+^/mm) (*n* = 4 animals). (**J**) Representative images of TRAP staining. Scale bars represent 50 μm. (**K**) Quantitative analysis of TRAP^+^ length per bone surface (*n* = 4 animals). Quantitative values are presented as the mean ± S.E. One-way ANOVA was used to conduct multiple comparisons, and the Bonferroni test was used to identify where the differences lay. * *p* < 0.05 compared with the HKSA group, ** *p* < 0.01 compared with the HKSA group, NS means no significant difference.

## Data Availability

All datasets generated for this study are included in the article.

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
