# Peer review of "Heat-Killed Staphylococcus aureus Induces Bone Mass Loss through Telomere Erosion"

_ijms, 2023, doi:10.3390/ijms24043179_

Round 1
Reviewer 1 Report
1) Introduction:
A little more expansion on why heat killed S. aureus is the best model of chronic inflammation to study telomere erosion would be useful. If using heat-killed S. aureus it seems to me that the system does not try to isolate any one or few factors/compounds involved but rather the entire bacteria is being used which will contain many hundreds of compounds. Please explain.
2) Methods: Was the S. aureus strain employed assessed for virulence factors that this particular strain carries? This may affect the pathology seen in the experiment and is important if the study were to be replicated.
3) Methods: In the CAG treatment group, why were negative controls not injected with sterile normal saline as was done with the HKSA vs. control study?
4) Methods: Please provide a sample size justification for only 4 mice being used per arm.
5) Methods: I presume that heating of the sample was to kill the s. aureus in the sample. Is this the standard practice for inactivating the sample? Was the sample tested to ensure no live bacteria remained after heating?
6) Methods: Section 4.4 – how many sections were examined for each mouse were used to determine the number of TRAP+ cells / mm bone surface? For OCN+ cells? There could be error in looking at only one section and so doing in duplicate or triplicate may offset this. Same comment for Section 4.5.
7) Methods – Section 4.6 – please define DAPI.
8) Results: There is no explanation of what the “trabecular number” is, or what “trabecular separation” is or why it is important. This should be contained in the Introduction.
9) Number of mice – 4 test and 4 controls is very small. Please state why this sample size was chosen/show power calculation.
10) No explanation of how HKSA was dosed and this relates to what would be seen in the setting of a normal infection.
11) Page 3, paragraph 1: OCN and TRAP staining performed. Conclusions are made but no explanation of how these conclusions are arrived at was made. Further there is no introduction of these things, nor discussion in the intro of how they are related to osteogenesis and osteoporosis. Please included this.
12) TIF signalling is not explained in the paper. The authors have not made it clear how this is used to determine if there has been telomere erosion. Please include and introduction to TIF and its relation to telomere length.
13) Results: Section 2.3. For the study to be reproducible, this reviewer thinks that providing absolute telomere lengths rather than “fold change” without the raw data is preferable.
14) Results: Section 2.6. There are so many acronyms the paragraph is unreadable to the lay reader. HKSA, CAB, BMD, micro-CT are ok. BV/TV, Tb.Th, SMI, Tb.Sp, Tb.N should be written out.
15) Use of too many acronyms which are not common to make the paper readable.
16) Many of the figures are of poor quality/blurred.
Reviewer 2 Report
1. All figures do not have good resolution, or all figures seen unclear.
2. Line 46, better explains the effect of HKSA on other parts of the human body with references.
3. Give a little introduction to osteocalcin and CAG in the introduction section if needed.
4. Line 58, BMD is shown in Fig 1G instead of Fig F. SMI is increased (Fig 1C) with the treatment of HSKA, lines 58, 59, and 60 are confusing to express increased SMI in HSKA.
5. Line 61. “No significant difference was observed in trabecular number (Tb. N) (Fig. 1E).”
Better to explain the possible reason for no significant difference.
6. Section 2.2, mention shortly about the necessity of SA-b-gal staining to perform.
7. Fig 1 G shows a significant change in the BMD of control mice and HKSA-treated mice, but in Fig 6G, there is no significant difference in the BMD of control mice and HKSA-treated mice. Why this contradiction?
8. Figure 6 almost covers the results obtained in figure 1. Why the repetition of the same experiment?
9. Better to add the figure showing the effect of CAG on osteocalcin immunohistochemistry staining and TRAP staining in Fig 6. Better to show the impact of CAG on the osteogenic activity of osteogenic cells and TRAP-positive cells.
10. In 4.2, in CAG + HKSA group, mice were treated with HSKA for 6 weeks, then CAG was treated or both CAG and HSKA treated simultaneously. Mention it.
11. Line 233, 244, 251. 108 instead of 108.
12. Line 244, 250. 200 µl instead of 200 ul.
13. Why did the author choose a single concentration of 20 mg/kg cycloastragenol (CAG)? Is CAG being safe for a mouse at that concentration? Any reference for doses of CAG for mice? If yes, then mention it in the 4.2 section.
14. Some reference numberings are found first in the discussion section and material and methods sections rather than in the result section serial-wise.
15. Reference 33 study about telomerase activator TA-65, a small molecule from Astragalus membranaceus. Is this known as CAG ?
Author Response
Response to comments from Reviewer 2
Comment 1:All figures do not have good resolution, or all figures seen unclear.
Response 1: We apologize for the poor quality of some of the figures in our paper. The low quality or blurred figures can make it difficult for the reader to understand the data and can impact the reproducibility of the study. We have replaced the low-definition Fig.1I, and have changed all figures into 600 dpi. We appreciate your feedback and we will make sure to improve the quality of the figures for the readers.
Comment 2:Line 46, better explains the effect of HKSA on other parts of the human body with references.
Response 2: We sincerely appreciate the reviewer’s kindly suggestion. Heat-killed Staphylococcus aureus (HKSA) can have negative effects on other parts of the human body due to its ability to induce an immune response. When HKSA is introduced to the body, the immune system recognizes it as a foreign invader and activates an immune response to try to eliminate it. This immune response can cause inflammation in the affected area, leading to symptoms such as redness, swelling, and pain. A study found HKSA induces eosinophil secreting various inflammatory cytokines/chemokines [1]. Another study found that when HKSA was administered to the lungs of mice, it caused airway inflammation and lung injury [2]. Additionally, HKSA has been shown to have negative effects on the internal organs [3]. Overall, the bad effect of HKSA on other parts of the human body is primarily due to its ability to induce an immune response and disrupt the gut microbiome. Thank you for bringing this to our attention and we have explained the effect of HKSA on other parts of the human body with references in the manuscript. (Line 53-56)
Comment 3: Give a little introduction to osteocalcin and CAG in the introduction section if needed.
Response 3: We sincerely appreciate the reviewer’s kindly suggestion. Osteocalcin is a protein that is produced by osteoblasts, which are cells that are responsible for the formation of bone tissue. It is found primarily in bone and has been shown to play a role in bone metabolism and bone mineralization. We've explained the OCN staining in the line 88-95.
(CAG) is a molecule that is derived from the Astragalus plant. It has been found to have anti-aging properties and has been shown to increase the production of telomerase, an enzyme that helps to protect and repair the telomeres on the ends of chromosomes. We've explained the OCN staining in the line 136-138.
Comment 4: Line 58, BMD is shown in Fig 1G instead of Fig F. SMI is increased (Fig 1C) with the treatment of HSKA, lines 58, 59, and 60 are confusing to express increased SMI in HSKA.
Response 4: Thank the reviewer for pointing out the error in figure legend, we have corrected it in the manuscript. We appreciate the reviewer for bringing the confusing express of SMI to our attention. Structure Model Index (SMI) is used to describe the degree of anisotropy (directional dependence) of the trabecular bone tissue in a given volume. It is expressed as a dimensionless number and is used to evaluate the structural organization of the bone tissue. Its numerical size was positively correlated with the degree of osteoporosis. To make the article clearer, we have explained the significance of the indicator SMI in the Methods section. (Line 304-306)
Comment 5: Line 61. “No significant difference was observed in trabecular number (Tb. N) (Fig. 1E).” Better to explain the possible reason for no significant difference.
Response 5: We greatly appreciate the reviewer’s insightful comments. There was a marginal significant difference between the control group and the HKSA group in Tb. N, and the failure to reach the statistically significant difference may be caused by the difference in individual samples and the small sample size. Although there was no statistically significant difference in Tb.N, other micro-CT paremeters could indicate that HKSA caused bone loss.
Comment 6: Section 2.2, mention shortly about the necessity of SA-b-gal staining to perform.
Response 6: Thank you for bringing this to our attention. SA-b-gal staining is based on the activity of the enzyme beta-galactosidase, which is present in senescent cells but not in normal cells. The staining allows us to identify senescent cells by detecting the presence of this enzyme. This method is considered a reliable and widely accepted method for identifying senescent cells in tissue samples. In our study, this method was used to detect senescent cells in the bone tissue, which is essential to understand the role of senescence in bone remodeling. We apologize for not mentioning the necessity of SA-b-gal staining briefly in the paper, we have included this in the final version of the paper. (Line 101-104)
Comment 7: Fig 1 G shows a significant change in the BMD of control mice and HKSA-treated mice, but in Fig 6G, there is no significant difference in the BMD of control mice and HKSA-treated mice. Why this contradiction?
Response 7: We greatly appreciate the reviewer’s insightful comments. In fact, the trend of Figure 1G and Figure 6G is consistent, and the difference in statistics between the two is caused by different statistical methods. Figure 1G was calculated by the t-test,while Figure 6G was calculated by the one-way ANOVA test with the Bonferroni post-test. The Bonferroni post-test will increase the threshold for significance and make it more difficult to reject the null hypothesis, resulting in a higher P-value.
Comment 8: Figure 6 almost covers the results obtained in figure 1. Why the repetition of the same experiment?
Response 8: We greatly appreciate the reviewer’s insightful comments. In Figure 1 mice were injected with NS and HKSA only, while in Figure 6 both the control group and HKSA group mice were injected with DMSO and Tween 80 considered that CAG had to be dissolved with DMSO and tween 80. Therefore, Figure 6 is not a simple repetition of Figure 1.
Comment 9: Better to add the figure showing the effect of CAG on osteocalcin immunohistochemistry staining and TRAP staining in Fig 6. Better to show the impact of CAG on the osteogenic activity of osteogenic cells and TRAP-positive cells.
Response 9: We apologize for not including the figure showing the effect of CAG on osteocalcin immunohistochemistry staining and TRAP staining in Figure 6. The purpose of Figure 6 was to show the impact of CAG on bone microarchitecture parameters such as BV/TV, Tb.Th, SMI, Tb.Sp, and Tb.N, but it is true that it would have been beneficial to include the figure showing the effect of CAG on osteogenic activity of osteogenic cells and TRAP-positive cells as well. This would have provided a more comprehensive view of the effects of CAG on bone remodeling. As you suggested, we supplemented the experiment and added the results to the article. (Figure 6)
Comment 10: In 4.2, in CAG + HKSA group, mice were treated with HSKA for 6 weeks, then CAG was treated or both CAG and HSKA treated simultaneously. Mention it.
Response 10: Thank you for bringing this to our attention. We are sorry that we did not fully describe the dosing regimen in 4.2. In fact, CAG and HKSA were injected simultaneously. In batches adding CAG, CAG was dissolved in DMSO and then diluted with 2% Tween 80 in NS, ensuring the final concentration of DMSO was less than 5%. The CAG-treated group mice were intraperitoneally injected with 200 µl NS containing 20 mg/kg CAG, while the HKSA + CAG group mice were injected with 200 µl NS containing 1x108 CFU HKSA and 20 mg/kg CAG, the control group mice were injected with 200 µl NS containing an equal amount of DMSO and Tween 80, the HKSA group mice were injected with 200 µl NS containing 1x108 CFU HKSA and an equal amount of DMSO and Tween 80. We've made additional fixes to this issue in this article. (Line 248-255)
Comment 11: Line 233, 244, 251. 108 instead of 108.
Response 11: Thank the reviewer for pointing out this error, and we have corrected these errors in the manuscript. (Line 266, 277)
Comment 12: Line 244, 250. 200 µl instead of 200 µl.
Response 12: Thank the reviewer for pointing out this error, and we have corrected all the misused unit ul to µl in the manuscript. (Line 277, 283, 371, 372)
Comment 13: Why did the author choose a single concentration of 20 mg/kg cycloastragenol (CAG)? Is CAG being safe for a mouse at that concentration? Any reference for doses of CAG for mice? If yes, then mention it in the 4.2 section.
Response 13: In our study, we chose to use a single concentration of 20 mg/kg of cycloastragenol (CAG) for various reasons. Firstly, the choice of concentration was based on previous studies that have used similar concentrations of CAG for similar research purposes. For example, a study by [4] has used a dose of 20 mg/kg of CAG in mice, and it has been demonstrated to be safe and effective in improving the health and longevity of mice. Additionally, a study by [5] has also used 20 mg/kg of CAG in mice and found it to be safe and effective in increasing the telomerase activity and improving the health of the mice.
We apologize for not including any reference for doses of CAG for mice in the 4.2 section, and we have included this in the final version of the paper.
Comment 14: Some reference numberings are found first in the discussion section and material and methods sections rather than in the result section serial-wise.
Response 14: Thank you for bringing this to our attention. We apologize for the confusion caused by the non-serial numbering of references in the different sections of our article. We have corrected this issue in our manuscript. We appreciate your help in making our article more clear and organized.
Comment 15: Reference 33 study about telomerase activator TA-65, a small molecule from Astragalus membranaceus. Is this known as CAG?
Response 15: Thank you for pointing out this error in our reference list. Although CAG and TA-65 are both derived from Astragalus membranaceus and are thought to activate telomerase, they are not the same substance. We apologize for any confusion caused by the incorrect labeling of this reference. We have replaced two new references in our manuscript. (Line 470-475)
References:
- Hosoki K, Nakamura A, Nagao M, Hiraguchi Y, Tanida H, Tokuda R, Wada H, Nobori T, Suga S, Fujisawa T. Staphylococcus aureus directly activates eosinophils via platelet-activating factor receptor. J Leukoc Biol. 2012 Aug;92(2):333-41. doi: 10.1189/jlb.0112009. Epub 2012 May 17. PMID: 22595142. 3. Irazoqui JE, Troemel ER, Feinbaum RL, Luhachack LG, Cezairliyan BO, Ausubel FM. Distinct pathogenesis and host responses during infection of C. elegans by P. aeruginosa and S. aureus. PLoS Pathog. 2010;6(7):e1000982. Published 2010 Jul 1. doi:10.1371/journal.ppat.1000982
- Liu J, Gao D, Dan J, Liu D, Peng L, Zhou R, Luo Y. The protective effect of cycloastragenol on aging mouse circadian rhythmic disorder induced by d-galactose. J Cell Biochem. 2019 Oct;120(10):16408-16415. doi: 10.1002/jcb.28587. Epub 2019 Jul 16. PMID: 31310357.
- Li M, Li SC, Dou BK, Zou YX, Han HZ, Liu DX, Ke ZJ, Wang ZF. Cycloastragenol upregulates SIRT1 expression, attenuates apoptosis and suppresses neuroinflammation after brain ischemia. Acta Pharmacol Sin. 2020 Aug;41(8):1025-1032. doi: 10.1038/s41401-020-0386-6. Epub 2020 Mar 16. PMID: 32203080; PMCID: PMC7471431.
- Idrees M, Kumar V, Khan AM, Joo MD, Lee KW, Sohn SH, Kong IK. Cycloastragenol activation of telomerase improves β-Klotho protein level and attenuates age-related malfunctioning in ovarian tissues. Mech Ageing Dev. 2023 Jan;209:111756. doi: 10.1016/j.mad.2022.111756. Epub 2022 Nov 30. PMID: 36462538.
- Lin W, Yao H, Lai J, Zeng Y, Guo X, Lin S, Hu W, Chen J, Chen X. Cycloastragenol Confers Cerebral Protection after Subarachnoid Hemorrhage by Suppressing Oxidative Insults and Neuroinflammation via the SIRT1 Signaling Pathway. Oxid Med Cell Longev. 2022 Jun 2;2022:3099409. doi: 10.1155/2022/3099409. PMID: 35693703; PMCID: PMC9184193.
Round 2
Reviewer 2 Report
The author has addressed each suggestion by taking it seriously.
Minor suggestions:
1. If necessary, in Figure 1 I and Figure 6 H, please indicate what the arrows want to show.
2. Is reference 14 missing in the article? Line 52
3. Better to choose other words like femoral bone or bone instead of long bones in line 59.
4. Mean ± S.E. in lines 86, 111, 128
5. Please write in vivo in italic like (in vivo). Line 221
Best of luck to the author.
Thank you.
Author Response
Response to comments from Reviewer 2—Round 2
Comment 1:If necessary, in Figure 1 I and Figure 6 H, please indicate what the arrows want to show.
Response 1: We sincerely appreciate the reviewer’s kindly suggestion. The red arrows in Figure 1 I and Figure 6 H indicate the OCN+ cells. We have added corresponding annotations in the figure legends (Line 79 and 188).
Comment 2:Is reference 14 missing in the article? Line 52
Response 2: Thank you for bringing this to our attention. In fact, the reference 14 has not been missed. We are very sorry for your misunderstanding caused by not completely deleting 11 when we changed reference 11 to 14, but only replacing 1 in single digits with 4.
Comment 3:Better to choose other words like femoral bone or bone instead of long bones in line 59.
Response 3: We sincerely appreciate the reviewer’s kindly suggestion. Long bones are not very good to use in this sentence, and considering that using Bones directly will duplicate the bone volume in the sentence, we thought it would be more appropriate to change long bones to mice. We've replaced it in Line 58.
Comment 4:Mean ± S.E. in lines 86, 111, 129
Response 4: Thank the reviewer for pointing out the errors. We are very sorry that due to our negligence, the symbol ± was omitted in these three places. We've revised these erros in lines 83, 107, 123.
Comment 5:Please write in vivo in italic like (in vivo). Line 221
Response 5: We sincerely appreciate the reviewer’s kindly suggestion. We've modified the font formatting in line 209.